# Sexual transmission of murine papillomavirus (MmuPV1) in *Mus musculus*

**Megan E Spurgeon, Paul F Lambert\***

McArdle Laboratory for Cancer Research, Department of Oncology, University of Wisconsin-Madison School of Medicine and Public Health, Madison, United States

**Abstract** Human papillomaviruses (HPVs) are the most common sexually transmitted infectious agents. Because of the species specificity of HPVs, study of their natural transmission in laboratory animals is not possible. The papillomavirus, MmuPV1, which infects laboratory mice (*Mus musculus*), can cause infections in the female cervicovaginal epithelium of immunocompetent mice that progress to cancer. Here, we provide evidence that MmuPV1 is sexually transmitted in unmanipulated, immunocompetent male and female mice. Female 'donor' mice experimentally infected with MmuPV1 in their lower reproductive tract were housed with unmanipulated male mice. The male mice were then transferred to cages holding 'recipient' female mice. One third of the female recipient mice acquired cervicovaginal infections. Prolonged infections were verified by histopathology and *in situ* hybridization analyses of both male and recipient female mice at the study endpoint. These findings indicate that MmuPV1 is a new model animal papillomavirus with which to study sexually transmission of papillomaviruses.
DOI: https://doi.org/10.7554/eLife.50056.001

## Introduction

Human papillomaviruses (HPVs) are the most common sexually transmitted infection in the United States (*Satterwhite et al., 2013*), and oncogenic high-risk HPVs alone account for approximately 5% of cancers worldwide (*zur Hausen, 2009*). Despite the significant public health burden caused by these small DNA tumor viruses, research on papillomavirus sexual transmission has been severely limited due to the paucity of small animal preclinical models resulting from strict virus species-specific tropism. Mucosal tissues of the female reproductive tracts of rabbits (*Harvey et al., 1998*), multimammate rats (*Nafz et al., 2007*; *Nafz et al., 2008*), and rhesus macaques (*Wood et al., 2007*) are susceptible to rabbit oral papillomavirus (ROPV), mastomys coucha papillomavirus 2 (McPv2), and rhesus macaques papillomavirus (RhPV), respectively. However, natural transmission of a papillomavirus through sexual contact has only been reported in rhesus macaques (*Ostrow et al., 1990*). Unfortunately, cost and lack of suitable molecular biology tools and reagents for these models have generally deterred their broad scale use (*Christensen et al., 2017*; *Uberoi and Lambert, 2017*).

The recent discovery of a murine papillomavirus (MmuPV1 or MusPV1) (*Ingle et al., 2011*) alleviates many of these limitations, allowing the study of papillomavirus infection and disease in laboratory mice, which are tractable, genetically modifiable, and relatively affordable. MmuPV1 infects and causes disease at both cutaneous and mucosal sites of several common strains of laboratory mice (*Uberoi and Lambert, 2017*; *Cladel et al., 2015*; *Cladel et al., 2017a*; *Cladel et al., 2013*; *Handisurya et al., 2014*; *Handisurya et al., 2013*; *Hu et al., 2015*; *Jiang et al., 2017*; *Joh et al., 2012*; *Sundberg et al., 2014*; *Wang et al., 2015*). We recently published that MmuPV1 infection of the female reproductive tract causes neoplastic disease in immunocompetent *FVB/N* mice (*Spurgeon et al., 2019*). The severity of disease is exacerbated by treatment with estrogen (E2)

**\*For correspondence:**
plambert@wisc.edu

**Competing interests:** The authors declare that no competing interests exist.

**eLife digest** Human papillomaviruses are responsible for about 5% of all cancers: infections with high-risk strains of the virus lead to the vast majority of cervical cancers, other cancers of the anus and genital area, as well as a growing fraction of head and neck cancers. In humans, these viruses are transmitted through sexual contact.

Other animals do not get infected by human papillomaviruses, and this makes it difficult to study in the laboratory how these viruses pass from one individual to another. However, a mouse papillomavirus has recently been identified: known as MmuPV1, it also causes cervical cancer in rodents, but it was unknown whether it was transmitted sexually.

To investigate this question, Spurgeon and Lambert experimentally infected female mice with MmuPV1 and allowed them to have intercourse with healthy males. When the males then were mated to healthy females, approximately a third of these female mice became infected with MmuPV1. Males that transmitted the virus were also found to have penile infections. These results show that, like the human papillomavirus, MmuPV1 spreads through sexual interactions.

Knowledge gathered by studying MmuPV1 could help to understand sexually transmitted human papillomaviruses that cause cancer. Additional work could look into how the virus leads to cancer and investigate the viral and host factors that contribute to sexual transmission. Further studies may also focus on testing drugs that prevent transmission or eliminate the persistent infections that can lead to cancer.

DOI: https://doi.org/10.7554/eLife.50056.002

alone or in combination with ultraviolet B radiation (UVB), which induces prolonged systemic immunosuppression (*Uberoi et al., 2016*), leading to precancerous lesions and squamous cell carcinoma (SCC). Here, we describe application of our MmuPV1 infection cervicovaginal model to study MmuPV1 sexual transmission. We report natural papillomavirus sexual transmission in immunocompetent, unmanipulated male and female mice.

## Results and discussion

### Rationale and experimental design for MmuPV1 sexual transmission studies

By 4 months following experimental infection with MmuPV1 in their lower reproductive tract and treatment with E2 and UVB, immunocompetent *FVB/N* female mice develop high-grade precancerous cervicovaginal lesions and SCCs (*Spurgeon et al., 2019*). These lesions were associated with highly productive MmuPV1 infections throughout the cervicovaginal epithelia, as evidenced by strongly positive immunohistochemical staining for the major viral capsid protein L1 within the female reproductive tract (*Spurgeon et al., 2019*) (see also *Figure 1A*). This observation prompted us to test whether MmuPV1 can be sexually transmitted. Cohorts of female mice (referred to as 'Donors') that were either mock-infected or experimentally infected with MmuPV1+UV+E2 were held for 4 months (*Figure 1B*). The female Donors were then used to establish monogamous breeding pairs with uninfected male mice (referred to as 'male Breeders') and breeding allowed for at least 3 weeks. Male Breeders were then transferred into a cage with an uninfected female mouse (referred to as a 'Recipient') for at least 3 weeks. While the female Donors were treated with medroxyprogesterone acetate (Depo-Provera) and nonoxynol-9 to potentiate MmuPV1 infection (*Spurgeon et al., 2019*; *Roberts et al., 2007*), it is important to emphasize that none of the male Breeders were experimentally manipulated prior to or during matings and the female Recipients were not experimentally manipulated unless indicated below. We performed four separate transmission experiments summarized in *Figure 1B* and *Table 1* using various conditions. In Experiments 1 and 2, breeding occurred for 3 weeks with both the Donor and Recipient, and Recipient female mice were treated with E2 for 2 months starting at 8 weeks post-breeding. In Experiment 3, a fraction of Recipients (n = 4) were pretreated with Depo-provera 5 days prior to breeding, and in Experiment 4, male Breeders remained with Donors and Recipients for 8 weeks each instead of 3 weeks. For Experiment 4, the Donors from Experiment 3 were used as the source of MmuPV1. All experiments

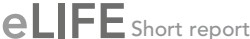

**Figure 1.** Rationale and experimental design for MmuPV1 sexual transmission studies. (**A**) A full-slide scan of a representative H and E-stained female reproductive tract from a Donor infected for 4 months with MmuPV1+UV +E2 with anatomical regions labeled. On the right, higher magnification images of the cervicovaginal fornix (inset) stained with H and E (left) or immunofluorescence for keratin (KRT; red) and MmuPV1 L1 capsid protein (L1;

*Figure 1 continued on next page*

*Figure 1 continued*

green). (**B**) Schematic of MmuPV1 sexual transmission experimental design. Mice infected or potentially infected are indicated in red. (**C**) DNA was isolated from cervicovaginal lavage samples from a group of representative MmuPV1+UV+E2-infected females that were used as Donors in Experiments 3 and 4. Lavages were conducted at the onset of Experiment 3 (4 months post-infection), the onset of Experiment 4 (8 months post-infection) and Experiment four endpoint (10 months post-infection. DNA was analyzed by PCR for the MmuPV1 E2 gene (top) or for the p53 gene (bottom) to verify DNA presence/quality.

DOI: https://doi.org/10.7554/eLife.50056.003

were conducted with wild-type *FVB/N* mice, totaling 9 mock-infected and 22 MmuPV1 Donor-positive breeding pairs. Prior to housing with male Breeders, we first assessed whether the female Donors harbored infections in their reproductive tracts by performing cervicovaginal lavage (CVL). DNA recovered from the CVLs were subjected to PCR to detect MmuPV1 DNA and the host gene, p53, as a positive control (*Figure 1C*). All female Donors were found to have MmuPV1 infections based upon the CVL/PCR tests. This confirmed our previously published results that MmuPV1+UV+E2-infected mice efficiently establish infections that persist for at least 4 months (*Spurgeon et al., 2019*). Indeed the infections of these female Donors persisted for up to 10 months post-infection (*Figure 1C*).

## Evidence for sexual transmission: Assessment of MmuPV1 infection status in female Recipient reproductive organs

To monitor for evidence for sexual transmission, we monitored the MmuPV1 infection status of the reproductive tracts of the female Recipient mice. CVL/PCR was performed on these mice starting approximately 3 weeks following introduction of the male Breeder mouse and approximately every month thereafter (*Figure 2A*). All female Recipient mice whose matings resulted in pregnancy were allowed to deliver offspring prior to their first CVL/PCR screen. Using this screening method, we identified 32% (n = 7/22) of female Recipient mice to harbor infections within their reproductive tracts (*Figure 2B*). These infections were observed across all four experiments (*Table 1*). Of the MmuPV1-positive female Recipient mice, 57% (n = 4/7) established prolonged MmuPV1 infections (MmuPV1 positive for at least 2 CVLs) while 43% (n = 3/7) had transient infections (MmuPV1 positive for only one CVL) (*Figure 2B*, *Table 1*). Prolonging the exposure of male Breeders to both the female Donors and female Recipients from 3 weeks to 8 weeks in Experiment 4 resulted in a higher percentage of MmuPV1-positive Recipients (50%; n = 3/6) than observed in Experiments 1 (33%; n = 1/3), 2 (33%; n = 2/6), or 3 and 3* (14%; n = 1/7). Preconditioning female recipient mice with Depo-Provera, a contraceptive drug, did not appear to influence susceptibility of mice to MmuPV1 infection (25% MmuPV1 positive: n = 1/4).

In a separate set of experiments, we determined that co-habitation of female mice experimentally infected with MmuPV1 in their reproductive tracts with uninfected female mice did not lead to transmission of MmuPV1 infections to the reproductive tracts of the latter mice based on negative CVL/PCR results (*Figure 2C*), consistent with the premise that the infections arising in reproductive tracts of Recipient females housed with male Breeders (*Figure 2A and B*) resulted from sexual activity between the males and females.

One obvious positive readout for sexual activity is pregnancy. Of the 7 Recipient females that acquired MmuPV1 infections in their reproductive tract, 6 became pregnant during the course of being housed with the male breeders (*Table 1*). The single Recipient female that did not become pregnant had been pre-treated with the contraceptive Depo-Provera (Mouse #16 in Experiment 3*). Pregnancy in the Donor females was less penetrant with 4 out of the 7 Donor female mice initially housed with these same male Breeders having become pregnant (*Table 1*). This lower penetrance likely reflects that the Donor Females were originally experimentally infected with MmuPV1 using a protocol that involves treatment with Depo-Provera, which can prevent estrus cycling for an extended period of time. We were interested in learning whether pups born to infected mums would acquire MmuPV1 infections. While our analysis was not exhaustive, we did not find evidence for MmuPV1 infections in the skin of several offspring of MmuPV1-positive Donor Females that we screened for E4 mRNA using *in situ* hybridization.

**Table 1.** Overview of MmuPV1 sexual transmission experiments and results

| Expt. | Experimental Conditions | Treatment of DONOR Female | DONOR pregnancy | Infection status of MALE BREEDER | Infection status of RECIPIENT female | RECIPIENT pregnancy (# if > 1) | # Positive CVL (total # of CVLs) in RECIPIENT Female |
|---|---|---|---|---|---|---|---|
| 1 | 3 weeks breeding; Recipients untreated prior to breeding, placed on E2 8 weeks after introduction of male. | No Virus #1 | Yes | Negative | Negative | Yes | 0 (4) |
| | | No Virus #2 | No | Negative | Negative | Yes | 0 (4) |
| | | No Virus #3 | Yes | Negative | Negative | Yes | 0 (4) |
| 3 | 3 weeks breeding; Recipients untreated prior to breeding. | No Virus #4 | Yes | Negative | Negative | Yes | 0 (3) |
| | | No Virus #5 | Yes | Negative | Negative | No | 0 (3) |
| | | No Virus #6 | Yes | Negative | Negative | No | 0 (3) |
| 4 | Prolonged Donor and Recipient breeding with male (8 weeks). Recipients untreated prior to breeding. | No Virus #7 | Yes | Negative | Negative | Yes (2) | 0 (4) |
| | | No Virus #8 | Yes | Negative | Negative | Yes (2) | 0 (4) |
| | | No Virus #9 | Yes | Negative | Negative | Yes (2) | 0 (4) |
| 1 | 3 weeks breeding, Recipients untreated prior to breeding, placed on E2 8 weeks after introduction of male. | MmuPV1+UV +E2 #1 | No | Positive | Negative | Yes | 0 (4) |
| | | MmuPV1+UV +E2 #2 | Yes | Negative | Negative | Yes | 0 (4) |
| | | MmuPV1+UV +E2 #3 | Yes | Positive | Positive (Prolonged) | Yes | 3 (4) |
| 2 | 3 weeks breeding, Recipients untreated prior to breeding, placed on E2 8 weeks after introduction of male. | MmuPV1+UV +E2 #4 | No | Negative | Negative | Yes | 0 (5) |
| | | MmuPV1+UV +E2 #5 | No | Positive | Positive (Transient) | Yes | 1 (5) |
| | | MmuPV1+UV +E2 #6 | No | Negative | Negative | Yes | 0 (5) |
| | | MmuPV1+UV +E2 #7 | No | Negative | Negative | Yes | 0 (5) |
| | | MmuPV1+UV +E2 #8 | No | Negative | Negative | Yes | 0 (5) |
| | | MmuPV1+UV +E2 #9 | Yes | Negative | Positive (Transient) | Yes | 1 (5) |
| 3 | 3 weeks breeding; Recipients untreated prior to breeding, not treated with E2. | MmuPV1+UV +E2 #10 | No | Positive | Negative | Yes | 0 (3) |
| | | MmuPV1+UV +E2 #11 | No | Negative | Negative | Yes (2) | 0 (3) |
| | | MmuPV1+UV +E2 #12 | Yes | Negative | Negative | Yes | 0 (3) |
| 3* | 3 weeks breeding; female Recipients treated with Depo-Provera 5d prior to breeding, not treated with E2. | MmuPV1+UV +E2 #13 | No | Negative | Negative | No | 0 (3) |
| | | MmuPV1+UV +E2 #14 | No | Negative | Negative | No | 0 (3) |
| | | MmuPV1+UV +E2 #15 | Yes | Negative | Negative | No | 0 (3) |
| | | MmuPV1+UV +E2 #16 | Yes | Negative | Positive (Prolonged) | No | 2 (3) |

*Table 1 continued on next page*

*Table 1 continued*

| Expt. | Experimental Conditions | Treatment of DONOR Female | DONOR pregnancy | Infection status of MALE BREEDER | Infection status of RECIPIENT female | RECIPIENT pregnancy (# if > 1) | # Positive CVL (total # of CVLs) in RECIPIENT Female |
|---|---|---|---|---|---|---|---|
| 4 | Prolonged Donor and Recipient breeding with male (8 weeks). Recipients untreated prior to breeding, not treated with E2. | MmuPV1+UV+E2 #17 | No | Negative | Negative | Yes (2) | 0 (4) |
| | | MmuPV1+UV+E2 #18 | No | Negative | Negative | Yes (2) | 0 (4) |
| | | MmuPV1+UV+E2 #19 | Yes | Negative | Positive (Prolonged) | Yes (3) | 3 (4) |
| | | MmuPV1+UV+E2 #20 | No | Negative | Negative | Yes | 0 (4) |
| | | MmuPV1+UV+E2 #21 | No | Negative | Positive (Transient) | Yes (3) | 1 (4) |
| | | MmuPV1+UV+E2 #22 | No | Negative | Positive (Prolonged) | Yes (2) | 2 (4) |

DOI: https://doi.org/10.7554/eLife.50056.004

To confirm that the MmuPV1-positive PCR results from the CVLs reflect persistent infections of the cervical/vaginal epithelium, we performed endpoint histopathological and MmuPV1-specific *in situ* hybridization (RNAscope) analyses on the reproductive tract of a female Recipient mouse (Recipient #3), which was MmuPV1-positive at the endpoint by CVL/PCR (*Figure 2A and D*). RNAscope used probes to detect viral transcripts containing the E4 region because that region is present in most early and late transcripts (*Xue et al., 2017*). Several discrete regions of epithelia were positive for MmuPV1 viral transcripts. These regions correlated with histopathological signs of MmuPV1 infection (*Spurgeon et al., 2019*), including disorganization of the stratified epithelium, areas of hyperkeratinization, karyomegaly, perinuclear halos similar to koilocytes, and condensed chromatin. We also observed evidence for a productive viral infection as indicated by cells staining positively for the viral capsid protein L1 by immunofluorescence (*Figure 2D*), albeit at levels of detection that are much lower than that afforded by RNAscope-based detection of viral transcripts. The infected regions of epithelia were pathologically scored as having low-grade or mild dysplasia. This particular mouse was MmuPV1-positive by 6 weeks post-breeding, and treated for 2 months with estrogen starting at 8 weeks post-breeding. Our previous results indicate that neoplastic disease worsens in MmuPV1 and MmuPV1+E2-infected mice upon extended duration (4 or 6 months) (*Spurgeon et al., 2019*). It is therefore possible that MmuPV1-infected female Recipients may develop moderate to high-grade disease or even SCC if the infection is allowed to proceed for a longer period of time. We analyzed the reproductive tracts of additional Recipient female mice that were positive for MmuPV1 by CVL/PCR at the endpoint and found them to have sites of infections based upon MmuPV1 E4-specific *in situ* hybridization (data not shown). These results confirm that sexual transmission of MmuPV1 can lead to persistent infections in the absence of genetic or environmental manipulation.

## Male Breeders harbor infections in their reproductive organs

Because many female Recipient mice contracted MmuPV1 infections of their reproductive organs after being housed with male Breeders (*Figure 2*), we evaluated the reproductive organs of the male Breeders for the presence of MmuPV1. Attempts to detect the MmuPV1 by lavage of the male genitalia were not successful (insufficient DNA was retrieved based upon an inability to detect mouse p53 DNA by PCR; data not shown). Therefore we resorted to *in situ* hybridization analysis of male reproductive organs obtained at the time of euthanasia. We identified several male Breeders with MmuPV1-positive foci of infection by RNAscope (*Figure 3A*). All foci of infections were detected in epithelia of the penis, including the glans epithelium, mump ridge groove, and prepuce (foreskin)/preputial space (*Phillips et al., 2015*; *Rodriguez et al., 2011*) (*Figure 3B and C*). Notably, many of these sites are anatomical locations infected by HPV in men (*Giuliano et al., 2007*). We also

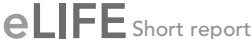

**Figure 2.** Evidence for sexual transmission: Assessment of MmuPV1 infection status in female Recipient reproductive organs. (A) DNA was isolated from cervicovaginal lavage (CVL) samples collected from a group of representative Recipient female mice at three different occasions. The numerical mouse identifiers correspond to mice listed in *Table 1*. The three different CVL time points are as follows (time is listed in weeks following introduction to male Breeder): Mock Recipient mice #1–3 and Recipients #1–3 from Experiment 1 (CVL1: 6 weeks, CVL2: 13 weeks, CVL3: 17 weeks), Recipient mice #5, #8, and #9 from Experiment 2 (CVL1: 8 weeks, CVL2: 11 weeks, CVL3: 13 weeks), Recipient mouse #16 from Experiment 3* (CVL1: 3 weeks, CVL2: 4.5 weeks, CVL3: 9 weeks), and Recipient mice #19, #21, and #22 from Experiment 4 (CVL1: 4.5 weeks, CVL2: 7 weeks, CVL3: 9 weeks). DNA was analyzed by PCR for the MmuPV1 E2 gene (top) or for the p53 gene (bottom) to verify DNA presence/quality. (B) Incidence of MmuPV1 infection via sexual transmission in Recipient females as determined by CVL for MmuPV1 E2 gene. (C) Schematic of co-habitation study in which each co-housed pair consisted of an experimentally MmuPV1-infected female mouse and an uninfected female mouse. After 3 weeks of co-habitation, DNA isolated from cervicovaginal lavages was analyzed by PCR for the MmuPV1 E2 gene (top) or for the p53 gene (bottom) to verify DNA presence/quality. (D) Full-slide scans of the female reproductive tract harvested from Recipient Mouse #3 with a prolonged MmuPV1 infection as a result of sexual contact. Tissue is stained with H

*Figure 2 continued on next page*

*Figure 2 continued*

and E (left) or for the MmuPV1 E4 viral transcript using RNAscope (right). Higher magnification images of the infected regions of epithelia are shown stained with H and E (top), RNAscope for the MmuPV1 E4 transcript (middle), and the MmuPV1 L1 protein (green) and keratin 14 (red) by immunofluorescence (bottom). White arrow indicates junction between uninfected and MmuPV1-infected epithelia. All scale bars = 100 μM.
DOI: https://doi.org/10.7554/eLife.50056.005

observed evidence for productive viral infections in the penis using L1 immunofluorescence (*Figure 3B*). Similar to our observation in the MmuPV1-positive Recipient females (*Figure 2D*), L1

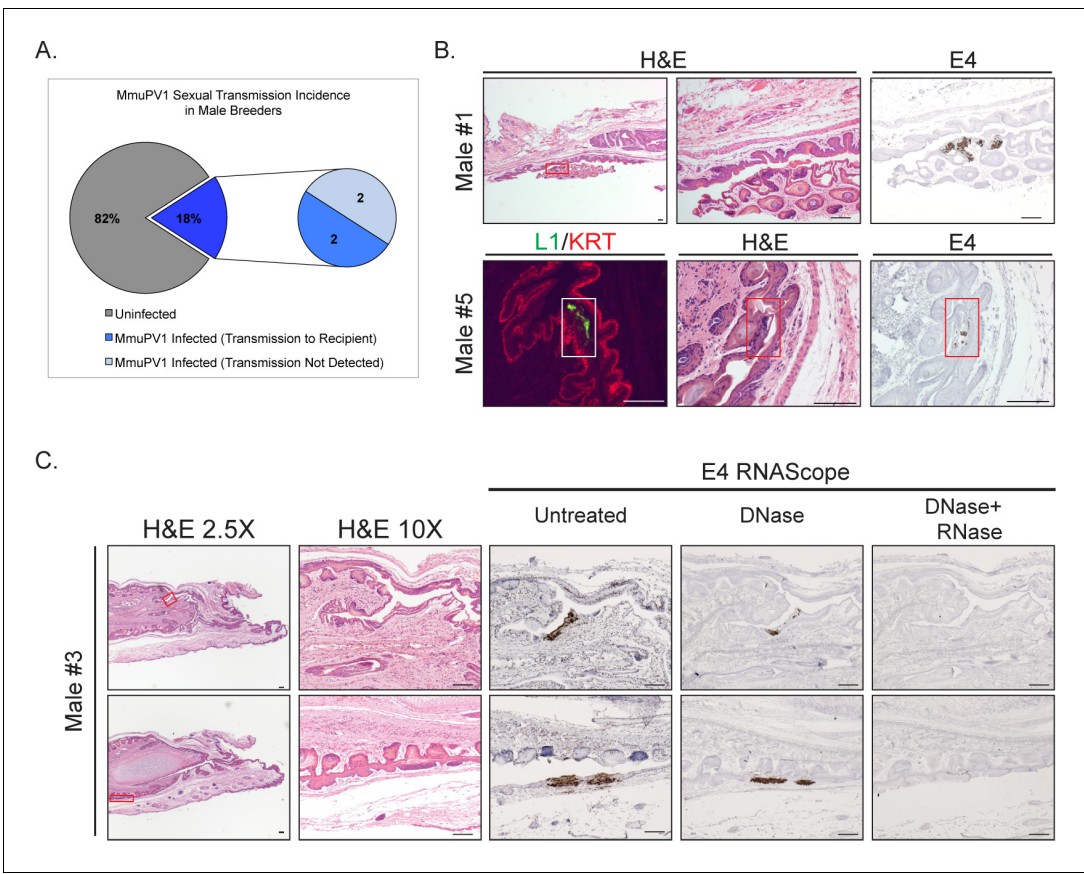

**Figure 3.** Male Breeders harbor infections in their reproductive organs. (**A**) Incidence of MmuPV1 infection via sexual transmission in male Breeders as determined by staining tissue for MmuPV1 E4 transcript using RNAscope. (**B**) Various regions of the penis in MmuPV1-positive male Breeders stained with H and E or RNAscope for MmuPV1 E4 viral transcripts. Lower magnification images on the left for Male #1 include inset boxes (red) indicating the region staining positive for E4, which is shown on the far right. Higher magnification H and E-stained image of the region is shown in the center. Top: Male #1 (did not transmit to Recipient) with MmuPV1-positive region in the glans penile epithelium (red inset). Bottom: Male #5 (transmitted to Recipient) with MmuPV1-positive region in the glans penile epithelium stained for L1 (green) and K14 (KRT; red) (white inset; left), H and E (red inset; middle), and E4 by RNAscope (red inset; right). (**C**) Tissue from Male #3 (transmitted to Recipient #3 shown in *Figure 2C*) is stained with H and E (left two columns) or for the MmuPV1 E4 viral transcript using RNAscope (right) three panels. Lower magnification images on the left include inset boxes (red) indicating the region staining positive for E4. Higher magnification images of the infected regions of epithelia are shown stained with H and E and RNAscope for the MmuPV1 E4 transcript (bottom). In the RNAscope analysis, slides were left untreated, treated with DNase to remove any signal from viral DNA, and with DNase+RNase to verify signal is specific for viral RNA transcripts. Top: Male #3 (transmitted to Recipient) with MmuPV1-positive region in the mump ridge groove of the glans penis (red inset). Bottom: Male #3 (transmitted to Recipient) with MmuPV1-positive region in the prepuce (foreskin)/inner preputial space (red inset). All scale bars = 100 μM.
DOI: https://doi.org/10.7554/eLife.50056.006

was detected albeit at a lower level than that observed for E4 transcripts, which uses the sensitive RNAscope technology. Two of the male Breeders (Male #3 and Male #5) that had detectable MmuPV1 infections on their penis by the endpoint RNAscope analysis were associated with MmuPV1-positive female Recipients (*Table 1*). The other two males that had detectable MmuPV1 infections on their penis did not appear to have transmitted MmuPV1 to Recipient females, based upon CVL/PCR. It remains possible that they did transmit, but the infections in the female Recipients were transient in nature and not caught by the intermittent CVL/PCR tests. Other males that were negative for MmuPV1-infections based upon endpoint RNAscope analysis did transmit MmuPV1 to female Recipients, suggesting they either had transient infections or else their foci of persistent infections were missed by the RNAscope analysis, which is very possible as only one section per male mouse was subjected to *in situ* hybridization. Of the 4 MmuPV1-positive male Breeders we identified using *in situ* hybridization, only 1 Donor-Breeder mating resulted in pregnancy, whereas all 4 Breeder-Recipient matings resulted in pregnancies (*Table 1*).

## Conclusions and significance

The data presented in this study provide strong evidence that MmuPV1 is sexually transmitted. MmuPV1 becomes the first model for studying sexual transmission of papillomaviruses in laboratory mice (*Mus musculus*). Our immediate goals are to use this natural sexual transmission model in immunocompetent mice to study the dynamics of sexual transmission, the role of host immunity, and methods for prevention and treatment.

## Materials and methods

### Key resources table

| Reagent type (species) or resource | Designation | Source or reference | Identifiers | Additional information |
|---|---|---|---|---|
| Strain, strain background | *FVB/N* | Taconic Biosciences | RRID:IMSR_TAC:fvb | Males (n = 31) Females (n = 31) |
| Strain, strain background | MmuPV1 | *Joh et al., 2012* *Uberoi et al., 2016* | GenBank: GU808564.1 | In-lab stock 'AU 11/13', pAU.4 |
| Antibody | Anti-MusPV1 L1 (rabbit polyclonal immune serum) | Chris Buck, NCI/NIH | | IF (1:5000) |
| Antibody | Anti-K14 (rabbit polyclonal) | BioLegend | Cat#905301; RRID:AB_2565048 | IF (1:1000) |
| Sequence-based reagent | MmuPV1_E2_1 | *Hu et al., 2015* *Cladel et al., 2017a* *Spurgeon et al., 2019* | PCR primers | GCCCGAAGACAACACCGCCACG |
| Sequenced-based reagent | MmuPV1_E2_2 | *Hu et al., 2015* *Cladel et al., 2017a* *Spurgeon et al., 2019* | PCR primers | CCTCCGCCTCGTCCCCAAATGG |
| Sequenced-based reagent | p53-1 | *Spurgeon et al., 2019* | PCR primers | TATACTCAGAGCCGGCCT |
| Sequenced-based reagent | p53-2 | *Spurgeon et al., 2019* | PCR primers | ACAGCGTGGTGGTACCTTAT |
| Sequenced-based reagent | p53-3 | *Spurgeon et al., 2019* | PCR primers | TCCTCGTGCTTTACGGTATC |
| Sequenced-based reagent | MusPV-E4 | *Xue et al., 2017* | RNAscope probe Cat#473281 | |
| Commercial assay or kit | RNAscope 2.5 HD Detection Kit Brown | ACDBio | Cat# 322300 | |
| Chemical compound, drug | Tyramide signal amplification (TSA)-related reagents | Online protocol: https://doi.org/10.17504/protocols.io.i8cchsw | | |

*Continued on next page*

*Continued*

| Reagent type (species) or resource | Designation | Source or reference | Identifiers | Additional information |
|---|---|---|---|---|
| Chemical compound, drug | medroxyprogesterone acetate | Amphastar Pharmaceuticals | Depo-Provera | 3 mg/animal, subcutaneous injection |
| Chemical compound, drug | Nonoxynol-9 (4%) | Options Conceptrol | Cat#247149 | 50 µl/mouse, intravaginal |
| Chemical compound, drug | Carboxyl methylcellulose (4%) | Sigma Aldrich | Cat# C4888 | 25 µl/mouse, intravaginal |
| Chemical compound, drug | 17β-estradiol pellet, 0.05 mg/60 days | Innovative Research of America | Cat#SE-121 | 0.05 mg/ 60 days, subcutaneous pellet |
| Other | Hematoxylin QS | Vector | Cat#H-3404 | Counterstain |
| Other | Shandon Instant Hematoxylin | Thermo Fisher | Cat#6765015 | H and E stain |
| Other | Eosin | Sigma Aldrich | Cat#E4382 | H and E stain |
| Other | DNase I | Thermo Fisher Scientific | Cat#EN0521 | 20 units/sample |
| Other | RNase A | Qiagen | Cat#1006657 | 500 µg/sample |
| Other | RNase T1 | Fermentas | Cat#EN0542 | 2000 units/sample |

## Animals

Immunocompetent, wild-type *FVB/N* mice (Taconic Biosciences; Albany, NY) mice were used in this study. All animal experiments were performed in full compliance with standards outlined in the 'Guide for the Care and Use of Laboratory Animals' by the Laboratory Animal Resources (LAR) as specified by the Animal Welfare Act (AWA) and Office of Laboratory Animal Welfare (OLAW) and approved by the Governing Board of the National Research Council (NRC). Mice were housed at McArdle Laboratory Animal Care Unit in strict accordance with guidelines approved by the Association for Assessment of Laboratory Animal Care (AALAC), at the University of Wisconsin Medical School. All protocols for animal work were approved by the University of Wisconsin Medical School Institutional Animal Care and Use Committee (IACUC, Protocol number: M005871).

## MmuPV1 cervicovaginal infection and treatment of Donor females

At 6–8 weeks of age, female virgin *FVB/N* mice were infected with MmuPV1 virus as described previously (*Spurgeon et al., 2019*). Briefly, mice were injected subcutaneously with 3 mg medroxyprogesterone acetate (Amphastar Pharmaceuticals, Rancho Cucamongo, CA) 4–7 days prior to MmuPV1 infection to induce diestrus. On the day of the infection, mice were pre-treated vaginally with 50 µL Conceptrol (Options, #247149) containing 4% nonoxynol-9 to induce chemical injury to the cervicovaginal epithelium. At 4 hr post-treatment with Contraceptrol, mice were infected intravaginally with $10^8$ VGE (viral genome equivalents) MmuPV1 virions suspended in 25 µL 4% carboxyl methylcellulose (Sigma, #C4888). The MmuPV1 virus stock used for infection was generated by isolating virions from papillomas arising on infected *FoxN1^{nu/nu}* mice. To treat mice with estrogen, a continuous-release estrogen (E2) tablet (17β-estradiol; 0.05 mg/60 days; Innovative Research of America, Sarasota, FL) was inserted subcutaneously in the shoulder fat pads of the dorsal skin. For those mice receiving estrogen, treatment began 5 days following MmuPV1 infection. A new tablet was inserted every 2 months as needed. Infection and estrogen treatment were performed while mice were anesthetized with 5% isoflurane. Animals were exposed to a single dose of UVB at 1000 mJ/cm$^2$. UVB was administered using a custom designed Research Irradiation Unit (Daavlin, Bryan, OH) with lamps controlled using Daavlin Flex Control Integrating Dosimeters.

## MmuPV1 sexual transmission

Donors were lavaged prior to breeding to confirm infection, and then introduced to male *FVB/N* to establish monogamous breeding pairs. After breeding with infected female Donor mice, the males were isolated for 2–5 days, then introduced to uninfected female Recipient mice. In all experiments

except Experiment 3, female recipient mice were 6–8 weeks old virgin mice that were not treated with depoprovera or nonoxynol-9. In Experiment 3, female Recipient mice were pre-treated with depoprovera 5 days prior to introduction of Male Breeder. Male breeding with female Recipient mice was allowed to proceed for 3 weeks in Experiments 1–3. In Experiment 4, breeding with female Donors and Recipients was allowed to proceed for 8 weeks.

## Vaginal lavage and detection of MmuPV1 by PCR

The method for detecting MmuPV1 DNA by PCR in vaginal lavages was modified from that described in Hu et. al. and Cladel et. al. (*Hu et al., 2015*; *Cladel et al., 2017b*). Briefly, 25 µL of sterile PBS was introduced intravaginally with a pipette tip, triturating 4–5 times prior to retrieval using the pipetteman. The vaginal lavages were stored at −20˚ C and DNA isolated using spin-columns (DNeasy Blood and Tissue Kit; Qiagen #69506, Hilden, Germany). Eluted DNA was analyzed by PCR using primers specific to the MmuPV1 E2 gene: MmuPV1_E2_1 (5′-GCCCGAAGACAACACCGC-CACG-3′) and MmuPV1_E2_2 (5′-CCTCCGCCTCGTCCCCAAATGG-3′) and analyzed using agarose gel electrophoresis. The presence of DNA suitable for PCR amplification was verified by performing PCR for the p53 gene. The primers for the p53 gene were as follows: p53-1 (5′-TATACTCA-GAGCCGGCCT-3′), p53-2 (5′-ACAGCGTGGTGGTACCTTAT-3′), and p53-3 (5′-TCCTCGTGC TTTACGGTATC-3′).

## Tissue procurement, Processing, and Histopathological Analysis

Reproductive organs were harvested, fixed in 4% paraformaldehyde, and paraffin-embedded. Serial sections (5 µm) were cut and every 10th section was stained with H and E and evaluated by histopathology. The scoring system for worst stage of disease has been described previously (*Spurgeon et al., 2019*).

## MmuPV1 L1- cytokeratin dual immunofluorescence and RNA *in situ* hybridization

A detailed protocol for detecting MmuPV1 L1 using a tyramide-based signal amplification (TSA) method is available at: dx.doi.org/10.17504/protocols.io.i8cchsw. MmuPV1 viral transcripts were detected using RNAscope 2.5 HD Assay-Brown (Advanced Cell Diagnostics, Newark, CA) according to manufacturer instructions with probes specific for MmuPV1 E4 (Catalog #473281) as described previously (*Spurgeon et al., 2019*; *Xue et al., 2017*). Tissue sections were treated following protease treatment and prior to probe hybridization with 20 units of DNase I (Thermo Fisher Scientific, #EN0521), or DNase I combined with 500 ug RNase A (Qiagen, #1006657) plus 2000 units RNase T1 (Fermentas, Waltham, MA, #EN0542) for 30 min at 40˚C. Slides were counterstained with hematoxylin before mounting and coverslipping.

## Image acquisition

High resolution wide-field fluorescent images were acquired using Leica SP8 3X STED microscope (*Xue et al., 2017*) by means of a 20X objective lens (Specifications: HC PL APO 20x/0.75 CS2, Dry) LAS-X suite (version: 2.0.1). Full slide scans of tissues were performed using Aperio ScanScope XT Slide Scanner using 20x/0.75 Plan Apo objective. All other images were captured using a Zeiss AxioImager M2 microscope and AxioVision software version 4.8.2 (Jena, Germany).

## Acknowledgements

We would like to thank Aayushi Uberoi for helpful and thoughtful discussions as we initiated this study. We thank members of RARC for providing animal care and the UWCCC Experimental Pathology Laboratory for processing tissue. We also thank Ella Ward-Shaw for embedding and cutting tissue, and for hematoxylin and eosin staining sections. We also thank Chris Buck (National Cancer Institute, Bethesda, MD) for providing us with antibody against MmuPV1 L1. We would also like to acknowledge Stephanie McGregor for consultation regarding pathology.

We thank the University of Wisconsin Translational Research Initiatives in Pathology laboratory, in part supported by the UW Department of Pathology and Laboratory Medicine and UWCCC grant

P30 CA014520, for use of its facilities and services. This work was supported by funding from the National Cancer Institute to PFL (R35CA210807, P01CA022443) and MES (R50CA211246).

## Additional information

### Funding

| Funder | Grant reference number | Author |
| --- | --- | --- |
| National Institutes of Health | CA022443 | Paul F Lambert |
| National Institutes of Health | CA210807 | Paul F Lambert |
| National Institutes of Health | CA211246 | Megan E Spurgeon |
| National Institutes of Health | CA014520 | Paul F Lambert |

The funders had no role in study design, data collection and interpretation, or the decision to submit the work for publication.

### Author contributions

Megan E Spurgeon, Paul F Lambert, Conceptualization, Resources, Data curation, Formal analysis, Supervision, Funding acquisition, Validation, Investigation, Methodology, Writing—original draft, Project administration, Writing—review and editing

### Author ORCIDs

Megan E Spurgeon (ID) https://orcid.org/0000-0002-9753-6615
Paul F Lambert (ID) https://orcid.org/0000-0001-7983-2755

### Ethics

Animal experimentation: This study was performed in strict accordance with the recommendations in the Guide for the Care and Use of Laboratory Animals of the National Institutes of Health. All of the animals were handled according to an approved institutional animal care and use committee (IACUC) protocol (#M005871) of the University of Wisconsin School of Medicine and Public Health.

### Decision letter and Author response

Decision letter https://doi.org/10.7554/eLife.50056.009
Author response https://doi.org/10.7554/eLife.50056.010

## Additional files

### Supplementary files

• Transparent reporting form
DOI: https://doi.org/10.7554/eLife.50056.007

### Data availability

All data generated and analyzed during this study are included in the manuscript and supporting files.

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
