## [Decision Letter]

Thank you for submitting your article "Sexual transmission of murine papillomavirus (MmuPV1) in Mus musculus" for consideration by *eLife*. Your article has been reviewed by three peer reviewers, and the evaluation has been overseen by a Reviewing Editor and Neil Ferguson as the Senior Editor. The following individuals involved in review of your submission have agreed to reveal their identity: Alison Anne McBride (Reviewer #2); Sam Campos (Reviewer #3).

The reviewers have discussed the reviews with one another and the Reviewing Editor has drafted this decision to help you prepare a revised submission.

The reviewers find the work novel, important, and acceptable with no additional experiments required, but they do ask you to address the points of clarification (including pregnancy) within the text and figures, as outlined in the reviews. We suggest (but do not require) addition of discussion and possible data regarding MmuPV1 serology.

Reviewer #1:

Demonstration of sexual transmission of a papillomavirus in a small animal model is a novel and important finding, especially considering that almost all of the HPV infections that initiate cancers are sexually transmitted. The finding can be considered rather unexpected since MnuPV was initially characterized as a cutaneous virus. The study is straightforward and limited in scope, but the primary conclusion is sufficiently supported by the data provided.

Minor Comments:

Several suggestions for modifying the text are as follows:

1) The Abstract should include some indication of the efficiency of transmission.

2) Subsection “Evidence for sexual transmission: Assessment of MmuPV1 infection status in female Recipient reproductive organs”, first paragraph: were any of the differences in transmission rates significant according to standard criteria?

3) Subsection “Male breeders harbor infections in their reproductive organs”: how long after initiating breeding with a donor female were these analyses undertaken?

4) Subsection “Male breeders harbor infections in their reproductive organs”: if histological sections are available, it would be interesting to evaluate the lesions for L1 protein by immunohistochemistry as an indication of whether or not the infections were productive.

5) It also would helpful to know if the breeding of breeder males with recipient females resulted in pregnancy, as in indication that pairs actually mated. This information seems particularly relevant to the two infected males than apparently didn't sexually transmit infections.

6) Subsection “MmuPV1 Cervicovaginal Infection and Treatment of Donor Females”: please indicate the source of the infectious virions. Also, some readers might benefit from defining "VGE" as "viral genome equivalents" here.

Reviewer #2:

Spurgeon and Lambert present a valuable study demonstrating sexual transmission of murine papillomavirus in mice. I believe that this is the first description of transmission of genital papillomavirus infection and so is of significant impact. The study is short, but straightforward and the results are convincing.

The figures are very nice, but Table 1 is very hard to follow. It is difficult to interpret whether each row is a mouse or group of mice, and whether the CVLs were performed several times on one mouse, or whether this number in parentheses represents the number of mice. It would also be better if the footnotes were incorporated into the table. I suggest generating a much more comprehensive table that could be presented as supplementary data and adding to the manuscript instead a figure with graphic timelines that describe the various treatment and breeding schedules. I think this would be much easier to follow.

Reviewer #3:

Spurgeon and Lambert present the first experimental sexual transmission of mouse papillomavirus in immunocompetent FVB/N mice. Female "donor" mice were experimentally infected with MmuPV1 (along with UVB/E2 treatment). These mice were cohoused (mated with) uninfected male breeders for at least 21 days. These males were then cohoused with uninfected female "recipients" for at least 21 days. Transmission of MmuPV1 infections (both transient and prolonged) from donors to both male breeder and female recipients was documented. This study is a landmark achievement, opening the possibility of experimental MmuPV1 transmission studies in a genetically tractable model organism.

In my opinion there are no substantive concerns, only minor suggestions/clarifications (see below).

Minor Comments:

1) Some mention and/or discussion of any resulting pregnancies is warranted. Did any of the females get pregnant? Were they allowed to birth pups prior to being sacrificed? If so were any of the pups screened for infection? If so what was the outcome?

2) The diagram in Figure 1B is a bit confusing. As drawn it looks like female donors are housed with males for 4 months.

3) The PCR data in Figure 2A should be labeled better- the three rows of PCR (either MmuPV1 or p53) correspond to three timepoints post-breeding. What are these timepoints? Can the figure be labeled appropriately? Or were these timepoints different for each of the breeding pairs/recipients?

---

## [Author Response]

Reviewer #1:[…] Minor Comments:Several suggestions for modifying the text are as follows:1) The Abstract should include some indication of the efficiency of transmission.

We state in the Abstract that: “One third of the female recipient mice acquired cervicovaginal infections”.

2) Subsection “Evidence for sexual transmission: Assessment of MmuPV1 infection status in female Recipient reproductive organs”, first paragraph: were any of the differences in transmission rates significant according to standard criteria?

We performed a two-sided Fisher’s exact test to compare transmission incidence between Experiment 4 (prolonged breeding with Donor and Recipient) and Experiment 1 (p=1.0), Experiment 2 (p=1.0), Experiment 3 (no pretreatment and Depo-Provera pretreatment combined; p=0.27), and Experiment 3* (only mice that received pre-treatment with DepoProvera) (p=0.57). As the p-values indicate, none of these comparisons reached statistical significance, and this is likely due to the small group sizes. Note in reviewing the text we realized that in the original submission we stated that the frequency of infection within Experiment 3 was 12.5% (i.e. 1 out of 8 Recipient Females infected). It was actually 14% (1 out of 7 mice). We made this correction to the text. The table accurately reflected the correct data.

3) Subsection “Male breeders harbor infections in their reproductive organs”: how long after initiating breeding with a donor female were these analyses undertaken?

In Experiments 1, 2, 3, and 3*, males were co-housed with the donor female and recipient female for 3 weeks each. In Experiment 4, housing of males with donor females, as well as males with recipient females, was extended to 8 weeks. This information is now listed as a column in the revised Table 1, as well as in a new graphical timeline in revised Figure 1B. CVL/PCR was conducted on female recipients at various timepoints after breeding, and the representative time points shown in Figure 2A (listed as times following introduction to the male breeder) are now indicated in the figure legend for Figure 2A.

4) Subsection “Male breeders harbor infections in their reproductive organs”: if histological sections are available, it would be interesting to evaluate the lesions for L1 protein by immunohistochemistry as an indication of whether or not the infections were productive.

We now included representative images of L1 immunofluorescence analysis in the revised Figure 2D for the female reproductive tract, and revised Figure 3B for the male reproductive tract. Overall, the amount of detectable L1 is lower than the level of detectable E4 transcript measured by RNAscope, reflective of the high sensitivity of the latter technique. We now refer to these results in the last paragraph of the subsection “Evidence for sexual transmission: Assessment of MmuPV1 infection status in female Recipient reproductive organs” and in the subsection “Male breeders harbor infections in their reproductive organs”.

5) It also would helpful to know if the breeding of breeder males with recipient females resulted in pregnancy, as in indication that pairs actually mated. This information seems particularly relevant to the two infected males than apparently didn't sexually transmit infections.

We agree with the reviewer that pregnancy data are valuable. We had collected that data and now provide it in the revised Table 1 and discuss there data in the subsection “Evidence for sexual transmission: Assessment of MmuPV1 infection status in female Recipient reproductive organs”, third paragraph and in the subsection “Male breeders harbor infections in their reproductive organs”. Pregnancies were common in Recipient females except those few that were treated with the contraceptive Depo-Provera. Pregnancies were less penetrant in the Donor females, which likely reflects that they had been treated with the same contraceptive at the time they were experimentally infected with MmuPV1. Depo-Provera is known to cause extended periods in which females fail to cycle and therefore would prevent pregnancies. Males were housed with Donor females starting 4 months post infection. Likely the contraceptive was only beginning to wane in its contraceptive properties by that time point.

6) Subsection “MmuPV1 Cervicovaginal Infection and Treatment of Donor Females”: please indicate the source of the infectious virions. Also, some readers might benefit from defining "VGE" as "viral genome equivalents" here.

We appreciate the reviewer suggesting this clarification. We have provided this information in the subsection “MmuPV1 Cervicovaginal Infection and Treatment of Donor Females”.

Reviewer #2:[…] The figures are very nice, but Table 1 is very hard to follow. It is difficult to interpret whether each row is a mouse or group of mice, and whether the CVLs were performed several times on one mouse, or whether this number in parentheses represents the number of mice. It would also be better if the footnotes were incorporated into the Table. I suggest generating a much more comprehensive table that could be presented as supplementary data and adding to the manuscript instead a figure with graphic timelines that describe the various treatment and breeding schedules. I think this would be much easier to follow.

We agree with the reviewer's concerns. We have revised Figure 1B to include a more complete graphical representation of the different experiments and revised the table to be more inclusive of information. We think the table remains a critical summary of data that is not summarized elsewhere in the manuscript and therefore should not be made into supplementary data.

Reviewer #3:[…] Minor Comments:1) Some mention and/or discussion of any resulting pregnancies is warranted. Did any of the females get pregnant? Were they allowed to birth pups prior to being sacrificed? If so were any of the pups screened for infection? If so what was the outcome?

Regarding pregnancies, please see response to reviewer 1 comment 5. Regarding whether pups acquired infections from their infected mums, we did look for this by screening the skin of some pups born to Donor females by *in situ* hybridization. We did not see any evidence for vertical transmission; however, we did not do an exhaustive survey. We make note of this in the third paragraph of the subsection “Evidence for sexual transmission: Assessment of MmuPV1 infection status in female Recipient reproductive organs.”.

2) The diagram in Figure 1B is a bit confusing. As drawn it looks like female donors are housed with males for 4 months.

We have modified Figure 1B to address the reviewer’s suggestion, as well as that of reviewer #2. We hope the modified figure more clearly conveys the experimental design.

3) The PCR data in Figure 2A should be labeled better- the three rows of PCR (either MmuPV1 or p53) correspond to three timepoints post-breeding. What are these timepoints? Can the figure be labeled appropriately? Or were these timepoints different for each of the breeding pairs/recipients?

The timepoints at which CVL/PCR was conducted were different for each of the recipients. This is in part because we tried to avoid performing CVL/PCR while females were overtly pregnant for fear that we would cause complications. Because we felt including this information as labels in the figure would be confusing and distracting, we now list this information in the figure legend for Figure 2A.